# Interaction of quercetin and epigallocatechin gallate (EGCG) aggregates with pancreatic lipase under simplified intestinal conditions

**Atma-Sol Bustos**[1,2]*, **Andreas Håkansson**[1], **Javier A. Linares-Pastén**[3], **J. Mauricio. Peñarrieta**[2], **Lars Nilsson**[1]

**1** Food Technology, Faculty of Engineering LTH, Lund University, Lund, Sweden, **2** School of Chemistry, Faculty of Pure and Natural Sciences, Universidad Mayor de San Andrés, La Paz, Bolivia, **3** Biotechnology, Faculty of Engineering LTH, Lund University, Lund, Sweden

* atma-sol.bustos@food.lth.se

**Data Availability Statement:** All relevant data are within the manuscript and its Supporting Information files.

## Abstract

Diets rich in flavonoids have been related with low obesity rates, which could be related with their potential to inhibit pancreatic lipase, the main enzyme of fat assimilation. Some flavonoids can aggregate in aqueous medium suggesting that the inhibition mechanism could occur on both molecular and colloidal levels. This study investigates the interaction of two flavonoid aggregates, quercetin and epigallocatechin gallate (EGCG), with pancreatic lipase under simplified intestinal conditions. The stability and the morphology of these flavonoid aggregates were studied in four different solutions: Control (water), salt, low lipase concentration and high lipase concentration. Particles were found by optical microscopy in almost all the solutions tested, except EGCG-control. The results show that the precipitation rate decreases for quercetin and increases for EGCG in salt solution and that lipase stabilize quercetin aggregates. In addition, both flavonoids were shown to precipitate together with pancreatic lipase resulting in a sequestering of the enzyme.

## 1. Introduction

Obesity has become one of the main health problems in the last decades, the number of people affected with it has nearly tripled between 1975 and 2016. According to the World Health Organization (WHO), 13% of the adult world population were obese in 2016 [1]. This disorder is one of the major risk factors for chronic diseases such as cardiovascular diseases, diabetes and some types of cancers [1, 2]. Obesity is defined as an excessive fat accumulation that may harm health. It is a problem that affects both high- and middle-income countries, principally in urban settings. The main cause of obesity is an energy imbalance between calories consumed and calories expended, where the excess of calories can be the result of a lipid rich diet [1]. The ingested lipids are primarily digested in the small intestine by pancreatic lipase, the main enzyme for lipids digestion [3]. The inhibition of this enzyme is one of the most common treatments against obesity and several compounds have been tested for this purpose. One example is tetrahydrolipstatin (Orlistat), one of the main anti-obesity drugs in the market that

**Funding:** A.B, M.P and L.N have received funding from Swedish International Development Agency (SIDA) [grant numbers 75000553-02] https://www.sida.se/English/ The funders had no role in study design, data collection and analysis, decision to publish, or preparation of the manuscript.

**Competing interests:** The authors have declared that no competing interests exist.

acts as competitive inhibitor binding covalently in the active site of lipase [4]. Nowadays, there is an increasing interest in providing foods that can aid in preventing obesity, in order to reduce the need for such pharmaceutical treatments in the future. A recent study has showed that flavonoid intake is inversely associated with obesity in women and men [5], suggesting that foods high in flavonoids could play a part in such diets.

Flavonoids, a group of bioactive compounds, are present in food from plant origin, such as apples, berries, citrus fruit, grapes, red wine and green tea [6, 7]. The intake of a diet rich in flavonoids is associated with lower risk of contracting several diseases, such as various types of cancer, obesity and cardiovascular disease [8, 9]. Some of these properties of the flavonoids have been related to their antioxidant properties [10], although other studies have suggested that flavonoids could have inhibitory effects on enzymes, as for instances digestive enzymes [11, 12]. Due to the fact that the metabolism of flavonoids occurs mainly in the intestine [13–15], the interaction between them and pancreatic lipase is relevant to study. Most mechanisms of action, proposed for flavonoids in the human body, are based on molecular interactions [9] and their interaction with pancreatic lipase are not an exception. For example, it has been reported that quercetin and epigallocatechin gallate (EGCG) act as pancreatic lipase inhibitors, binding to the enzyme near the active site [16, 17]. Nevertheless, findings have suggested that flavonoids can form aggregates and that they are able to inhibit several unrelated enzymes [18, 19]. McGovern et al. (2003), have proposed that flavonoids can reversibly sequester some enzymes instead of binding to them in specific sites [20]. Although this suggested mechanism has been observed for different aggregates, there are no studies investigating this mechanism for the interaction between flavonoid aggregates and pancreatic lipase.

Our previous study suggests that the presence of flavonoid aggregates affects the reproducibility of pancreatic lipase assays [18], but their mechanism of action as aggregates is still unknown. Since it is likely that the ingested flavonoids are present in the human digestive track as aggregates, we hypothesize that the interaction with pancreatic lipase follows an additional mechanism besides molecular inhibition. If this is so, the understanding of these aggregates under simulated intestinal conditions can help to clarify the mechanism of action between flavonoids aggregates and pancreatic lipase. For that reason, the present study focusses on studying the interaction of two well-known flavonoids and pancreatic lipase under simplified intestinal conditions. Based on our previous study, we chose quercetin and EGCG, two flavonoids found in teas and fruits in high concentrations, and that can form aggregates in aqueous solutions [18, 19, 21]

## 2. Material and methods

### 2.1. Chemical compounds

Pancreatic lipase from porcine pancreas Type II (30–90 units/mg protein using triacetin), EGCG (PubChem CID: 65064), quercetin hydrate (PubChem CID: 16212154), bovine serum albumin (BSA) and Trizma base (PubChem CID: 6503) were purchased from Sigma Aldrich (St.Louis, MO, USA). Dimethyl sulphoxide (PubChem CID: 679) was purchased from VWR Chemicals (Fontenay-sous-Bois, France and Leuve, Belgium, respectively). All chemicals had purity >95%.

### 2.2. Sample preparation

Samples were prepared based on the standard static *in vitro* digestion INFOGEST method [22]. Three stock solutions were prepared to give the specific sample conditions: 1) Simulated intestinal fluid (SIF) stock solution following the description in the INFOGEST method [22], 2) $CaCl_2$ 3 mM solution (this solution is part of the digestion system, but is added separately to

**Table 1. Sample composition.**

| Experimental conditions | Water (μL) | SIF stock solution (μL) | CaCl₂ solution (μL) | Saturated lipase solution (μL) | Flavonoid solution in DMSO (μL) |
|---|---|---|---|---|---|
| Control | 195 | - | - | - | 5 |
| Salt solution | 95 | 80 | 20 | - | 5 |
| Low lipase concentration | 95 | 55 | 20 | 25 | 5 |
| High lipase concentration | 95 | 0 | 20 | 80 | 5 |

avoid precipitation, see INFOGEST method for more details), 3) saturated lipase solution dissolving 100 mg of pancreatic lipase in 10 mL of SIF stock solution where the supernatant was collected and separated from the insoluble fraction by centrifugation at room temperature for 10 minutes at 11 000 x g. The final lipase concentration of the stock solution was 20 μM, determined by a BCA protein assay Kit (Thermo Scientific) using BSA as protein reference. In addition, nine different solutions of quercetin and EGCG were prepared in DMSO in order to get the following concentrations in the final solutions (Table 1): 40, 60, 80, 100, 200, 400, 600, 800, 1000 μM. Four different samples were prepared for each concentration of quercetin and EGCG, see Table 1. The samples were based in a "typical example" proposed by INFOGEST method for the intestinal phase, where some of the solutions were replaced by water in order to investigate mainly lipase interaction. The total volume for each solution was 200 μL.

## 2.3. Stability in solution

The stability of the different quercetin and EGCG samples were monitored by turbidity measurements, where an increase of turbidity reflects aggregate formation and loss of stability. For this purpose, the samples were incubated at 37 ºC and their changes in optical density were measured with a spectrophotometric microplate reader (Spectrostar Nano supplied with MARS data analysis software version 3.20 R2, BMG Labtech, Germany) at a wavelength of 800 nm for 2 hours (digestion time suggested by INFOGEST method) at 3 min intervals. The results are expressed as optical density (OD). A blank was measured for all the samples and sample analysis was performed in (at least) triplicate.

## 2.4. Morphology

Microscopy was performed with an Optika b 350 optical microscope and Optika Vision Pro 2.7 software (Optika, Italy) at 400x magnification. Only the highest concentration (1000 μM) was studied for the two respective flavonoid samples and at least duplicates were performed for each sample.

## 2.5. Capability of sequester lipase

**2.5.1. AF4 instrumentation.** An asymmetric flow field-flow fractionation (AF4) [23] method was applied to quantify the remnant lipase after its interaction with different concentrations of quercetin and EGCG.

The AF4 instrument used in this study was an Eclipse 3 + Separation System (Wyatt Technology, Dernbach, Germany) coupled to a multi-angle light scattering (MALS) detector (Dawn Heleos II, Wyatt Technology) with a wavelength of 663.8 nm and a UV detector operating at 280 nm (UV-975 detector, Jasco Corp., Tokyo, Japan). Both detectors were connected to the separation system. The carrier liquid was delivered using an Agilent 1100 series isocratic pump connected to a vacuum degasser (Agilent Technologies, Waldbronn, Germany). The injection onto AF4 channel was by an auto sampler Agilent 1100 series. The AF4 channel was a trapezoidal long channel (Wyatt Technology) with tip-to-tip length of 26.0 cm and inlet and

outlet widths of 2.15 and 0.6 cm, respectively and with a nominal thickness of 350 μm. The accumulation wall was formed by an ultrafiltration membrane of regenerated cellulose with a nominal cut-off of 10 kDa (Merck Millipore, Bedford, MA, USA). A bovine serum albumin (BSA) solution was used for checking that separation occurred in the AF4 system and to normalize the MALS detector.

**2.5.2. AF4 sample preparation.** Nine different concentrations of "high lipase concentration" samples were prepared as described above (see Table 1), 5 for quercetin: 60, 200, 400, 600 and 1000 μM and 4 for EGCG: 60, 200, 600 and 1000 μM. Before the injection, all samples were incubated for 2h at room temperature and then centrifuged at 11 000 x g for 10 min. The supernatant was collected and filtered with a syringe filter with 0.02 μm pore size (inorganic membrane filter Anotop, Whatman, Germany) in order to determine the remnant lipase content in solution. A sample containing pancreatic lipase but not quercetin or EGCG was used as a positive control. Duplicate analysis and blanks were done for all concentrations and the positive control.

**2.5.3. AF4 parameters.** The AF4 carrier liquid was a tris-HCl buffer solution (20 mM, pH 8).

A sample volume of 80 μL was injected onto the channel at a flow rate of 0.2ml/min in focusing mode for 2 min, followed by 3 min of focusing prior to elution. The elution was carried out at a constant cross flow of 6.5 ml/min for 20 minutes and flushed without cross flow for 7 minutes before the next injection. The detector flow in all the steps was 0.5 ml/min.

**2.5.4. AF4 data processing.** The data analysis was performed using Astra software 6.1 (Wyatt Technology). The molecular weight of lipase in the positive control was determined by the Zimm model [24] with a 1st order fit using 14 scattering angles, from 34.8˚ to 163.3˚. The UV extinction coefficient used was 1.4 ml/(mg cm) and a refractive index increment (dn/dc) of 0.185 ml/mg [25]. As the dn/dc used is for a generic protein (BSA), the molecular weights reported for lipase are somewhat apparent. The second virial coefficient was considered negligible.

Once the lipase peak was identified, its retention time was used as reference for the rest of the samples, where only the UV detector was taken into consideration in order to calculate the free lipase in solution after its interaction with the respective flavonoid. For this, peak maximum was used for relative concentration that was normalized with respect to the control sample (containing only lipase).

## 2.6. Statistical analysis

Two-sample t- tests were performed assuming unequal variance in the two samples. The significance limit was set to 1%.

## 3. Results and discussion

### 3.1. Stability in solution

In order to understand the aggregation behavior of the selected flavonoids, the optical density over time was measured after mixing the flavonoids with different solutions. Typical plots, obtained from quercetin and EGCG are presented in Fig 1, both of them are from low concentration lipase samples at flavonoid concentrations of 1000 μM each and plotted after blank corrections. The error bars represent the standard deviation of at least triplicate analysis. The EGCG values in the $y$ axis were multiplied by 7 in order to improve the visualization of the plot.

A change in optical density occurs when particles precipitate and cause turbidity. To obtain the precipitation rate, a linear regression analysis was carried out and the slope plotted as a

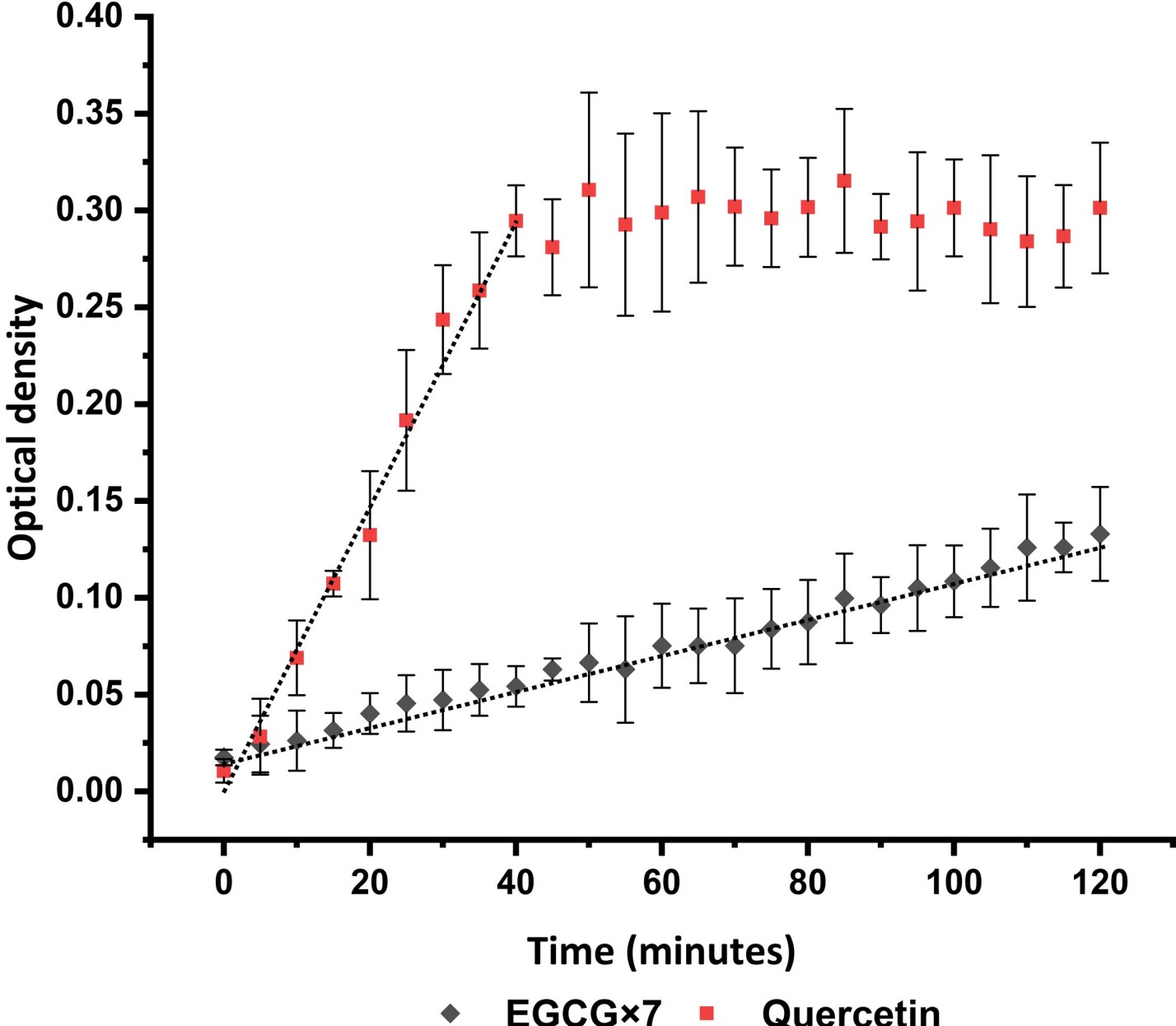

**Fig 1. Example of optical density over time curves of quercetin and EGCG (multiplied by 7 for visual purposes).** Concentrations of 1000 μM for each flavonoid prepared in the low lipase solution. The error bars are the standard deviation of at least triplicate analysis and the grey line represent the linear regression.

function of the flavonoid concentration, see (Fig 2A and 2B). For curves that reach a plateau (such as the quercetin in Fig 1), the regression was based on all data up to the time when it was judged that the plateau value was approximately reached, e.g., 40 min and optical density of 0.295 for the examples in Fig 1. This will result in a representation of the initial precipitation rate.

(Fig 2A and 2B) shows the initial precipitation rate of different quercetin and EGCG solutions. The Control sample for quercetin displays the initial precipitation rate at a flavonoid concentration $\geq 200$ μM. In salt solution, the onset of precipitation occurs at a higher

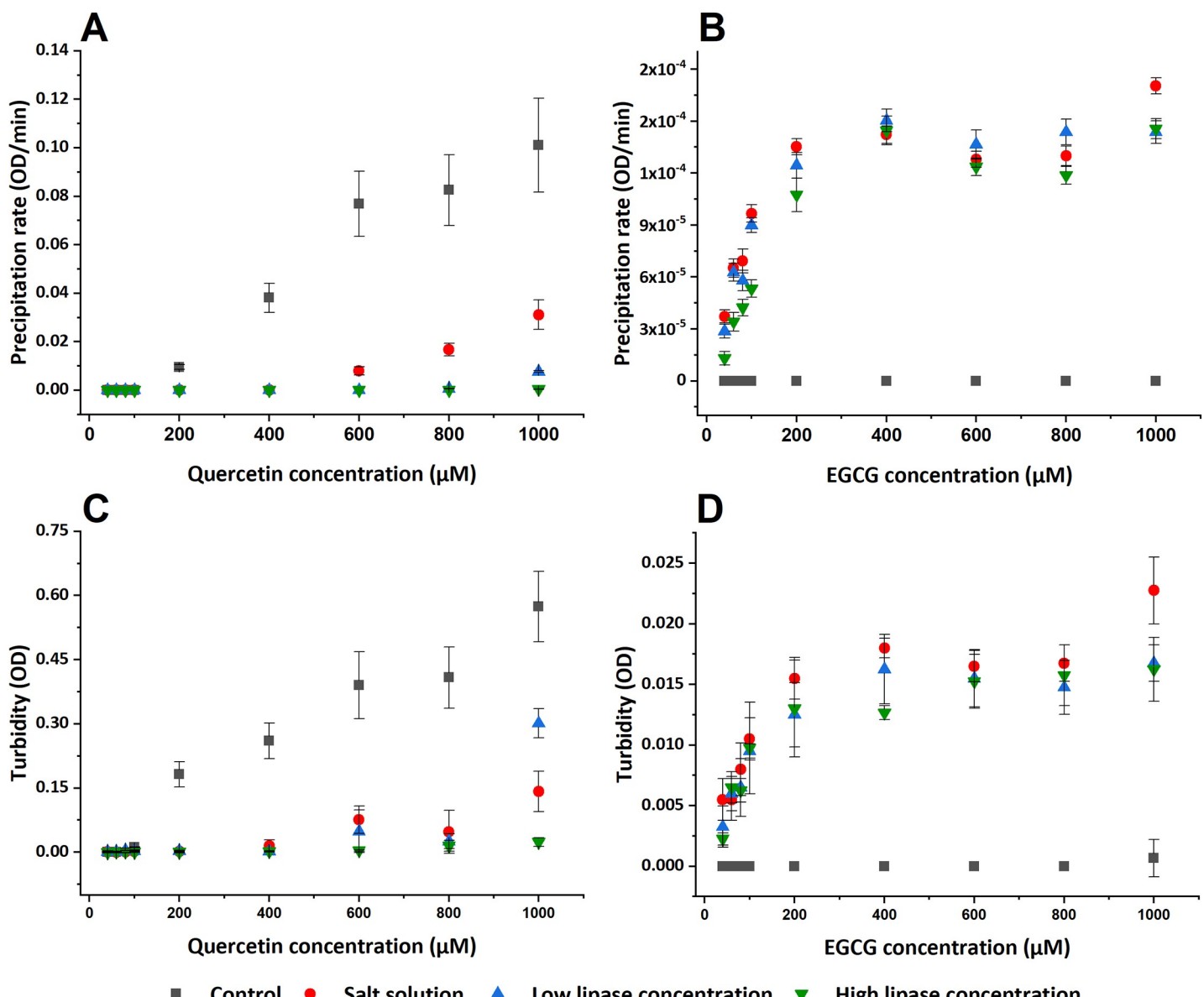

**Fig 2. Stability results.** (A, B) Initial precipitation rate of quercetin and EGCG, respectively. (C, D) Final turbidity results of quercetin and EGCG after 2 hours of incubation. The optical density (OD) was measured at a wavelength of 600 nm during 2 hours of incubation at 37˚C. The error bars represent the standard deviation of at least triplicates.

quercetin concentration (600 μM) and at a lower rate compared to the control sample. Quercetin in low lipase concentration only shows an initial precipitation rate at the highest flavonoid concentration tested (1000 μM), while it does not show any precipitation rate in high lipase concentration. In contrast, no precipitation can be observed for EGCG in the control sample. However, in the other three solutions, all the concentrations precipitate without any significant difference among them. In brief, the initial precipitation rate of EGCG samples is as follows: Control sample precipitates slower than salt solution and this is approximately equal to the rate in low and high lipase concentration samples, without significant differences among them. The rate of precipitation in the quercetin samples descends as follows: control, salt solution, low lipase and high lipase concentration. A high precipitation rate corresponds to a low

stability in solution. Thus, EGCG and quercetin display different behaviors in the investigated salt solution. EGCG that seems to be stable in the control sample, precipitates in salt solution, while the precipitation rate for quercetin decrease in the same solution. This could be related with the capability of quercetin to chelate various metal ions, such as $Ca^{+2}$ and $Mg^{+2}$ (ions present in SIF stock solution). The formation of these complexes normally occurs for flavonoids like quercetin and not in catechins, like EGCG [26–28]. More studies are needed to be conclusive, however this is outside of the scope of the current investigation.

In addition, lipase can also stabilize quercetin aggregates. The stabilization of aggregates by adsorbing proteins is well-known. Proteins are, in general, intrinsically surface active and when adsorbed they can give rise to steric and/or electrostatic repulsion that stabilize the suspension. Hence, it is possible that lipase adsorbed to the surface of the flavonoid particles exerts a stabilizing effect, resulting in the formation of flavonoid-lipase aggregates.

To have a general idea of the steady-state aggregates formed after precipitation, the final optical density obtained after 2 hours of incubation (corresponding to the time suggested by the INFOGEST method for the intestinal phase [22]) was plotted against the flavonoid concentration as shown in (Fig 2C and 2D).

Although the initial precipitation rate for quercetin in high lipase solution and EGCG in the control solutions are negligible (Fig 2(A)), the final optical density observed after 2 hours of incubation shows a significant optical density for some of the samples (see Fig 2(C)). This is more noticeable for the highest flavonoid concentration tested (see Table 2).

Turbidity is proportional to the diameter squared of the particles and directly proportional to their concentration [29]. Hence an increase in optical density could give an estimate of particle growth. The values reported in Table 2 suggest that the size of quercetin aggregates in the control are larger than in the salt solution and that the addition of a low lipase concentration increase its size, while high lipase concentration decrease it. On the other hand, EGCG aggregates in the control sample are smaller compared with the aggregates in other three samples (salt, low lipase and high lipase concentration). These last three samples do not present any significant difference among them. Table 2 also shows that quercetin aggregates are much larger than EGCG aggregates, two orders of magnitude higher in optical density.

## 3.2. Morphology

Optical microscopy was performed on all samples in Table 2 in order to study the morphology of the aggregates. Only the highest flavonoid concentrations in each solution was analyzed because they show a significant variability among them and the blank. Fig 3 shows microscope images for 1000 μM of quercetin and EGCG in the 4 different solutions investigated in this study.

Quercetin aggregates in the control solution show a jagged surface with a longitudinal diameter of approx. 1200 μm. These are also the largest aggregates compared to the other

**Table 2. Final optical density.**

| Sample | Quercetin (1000 μM) | EGCG (1000 μM) $\times 10^2$ |
|---|---|---|
| Control | 0.57±0.08 | 0.07±0.15 |
| Ionic | 0.14±0.05 | 2.3±0.28 |
| Low lipase concentration | 0.30±0.03 | 1.70±0.15 |
| High lipase concentration | 0.02±0.01 | 1.62±0.26 |

Optical density after two hours of incubation at 37˚C. The results are expressed as the average of at least triplicates ± one standard deviation.

## Quercetin

## EGCG

**Fig 3. Representative optical microscopy images of Quercetin and EGCG aggregates in different solutions.** A: Control, B: Salt solution, C: Low lipase concentration and D: High lipase concentration. Magnifications of 400 X for all the images. 1000 μM of flavonoid concentration in each sample.

investigated solutions. In the salt solution, quercetin aggregates are smaller than in the control, but keep the jagged surface and their shape tend to be spherical. The addition of a low concentration of lipase does not affect the size of the particles noticeably, but the morphology of the surface appears to change, making them appear smoother (see Fig 3 –Quercetin B). On the other hand, a high concentration of lipase reduces the particle size. These results follow a similar trend observed in the stability results (Table 2), where the optical density values descend in this order: Control, low lipase concentration, salt solution and high lipase concentration. In the case of EGCG, no particles were detected in the control sample. In the other three solutions, particles were also found, but without any notable difference among them, which is in agreement with the results from stability (see Table 2).

The results from stability and morphology sections confirm that quercetin and EGCG have poor solubility in aqueous solutions, as reported before [18, 21].

## 3.3. Capability of sequestering lipase

From the previous sections it was observed that lipase interacts with quercetin and EGCG aggregates, but to understand if these aggregates precipitated together with lipase or not, further investigations were undertaken. For this, AF4 was applied to quantify the lipase remaining in solution after its interaction with different concentrations of quercetin and EGCG. Any large aggregates were removed by filtration before AF4 analysis (see section 2.4.2 AF4 sample preparation).

The characteristic MALS and UV fractograms of pure lipase are shown in Fig 4(A), where two peaks were observed. The first peak, at elution time of 3.3 minutes, does not have light scattering (LS) signal, therefore its molecular weight was not determined. On the other hand, the second peak shows a molecular weight of 45 kDa at an elution time of 5.8 minutes. According to the SDS-PAGE, lipase have a molecular weight of ~50 kDa (S1 Fig), consistent with the value reported for commercial porcine pancreatic lipase [30]. This suggests that the molecular weight reported for the second peak correspond to the monomer of pancreatic lipase. Thus, the first peak could correspond to impurities present in the sample (S1 Fig) and will not be taken into consideration for the calculations.

Once the retention time for lipase was determined, the relative UV absorption was measured of the remnant lipase in all the sample of quercetin and EGCG, see (Fig 4C and 4D).

No additional peaks appear after the lipase solution has been mixed with quercetin (Fig 4 (C)). However, for EGCG an additional peak appears at retention time of 3 min (Fig 4(D)). This peak was present in all the blanks (EGCG samples without lipase), but does not interfere with the lipase peak quantification, when using the peak height as a relative absorption.

The relative UV absorption for each sample (relative to the positive control) was plotted as function of flavonoid concentration in Fig 4(B). The results show that the lipase concentration decrease when the concentration of flavonoids increases. Quercetin can sequester up to 50% of lipase under the simplified intestinal conditions tested, while EGCG only reduces it by 30% in the same conditions, showing that both flavonoids tested in this study precipitate together with lipase. This suggests that the interaction between pancreatic lipase and quercetin or EGCG aggregates gives rise to a sequestering mechanism. As mentioned in Section 1, this mechanism has been previously proposed by McGovern et al., where they indicate that some aggregates (including quercetin) can reversible sequester lactamase enzyme [31].

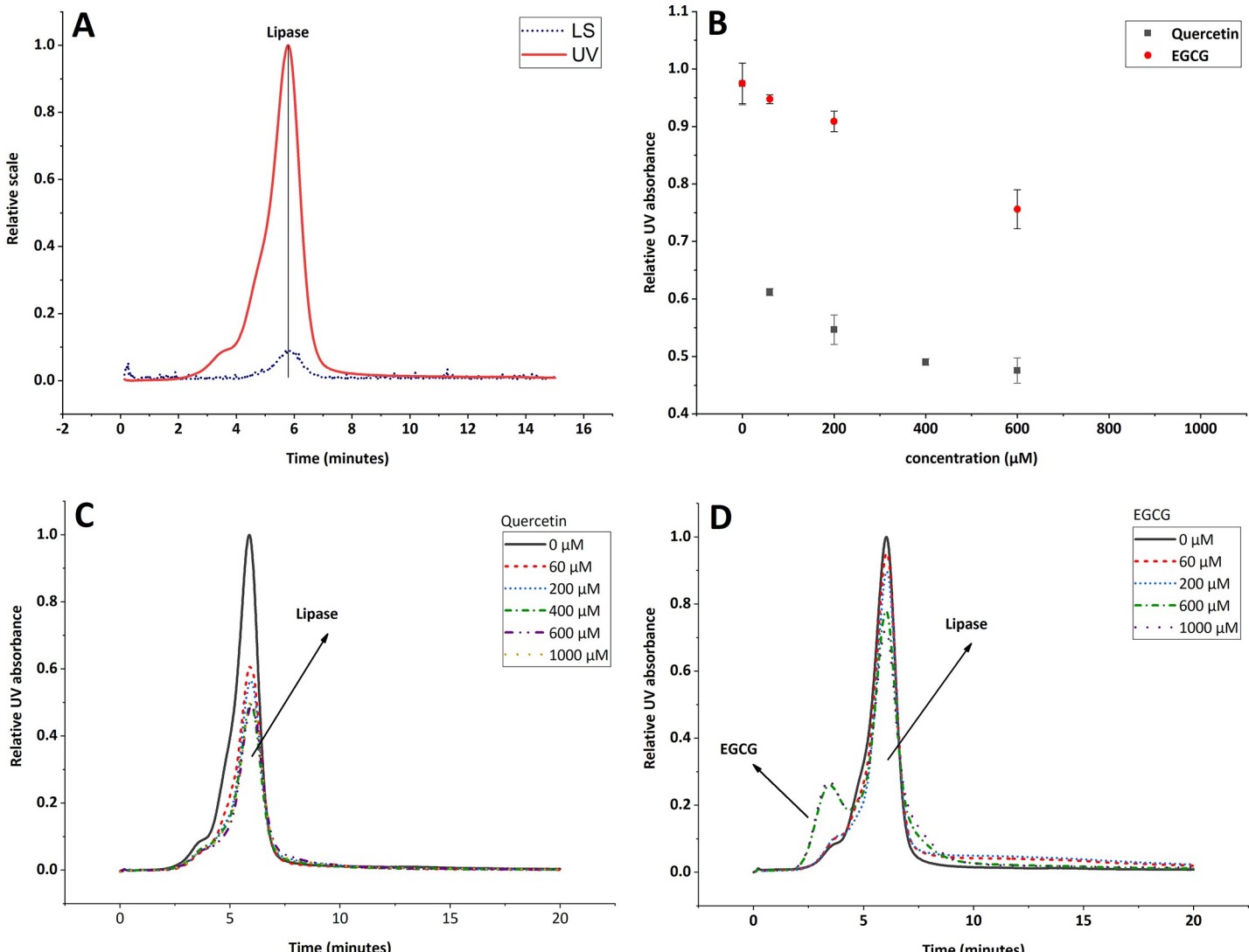

**Fig 4. Interaction of quercetin and EGCG aggregates with lipase, AF4 results.** (A) AF4-MALS-UV fractograms of pancreatic lipase. 40 µg of injected mass. LS is the light scattering signal at 90˚, UV is the UV-signal at 280 nm. (B) Remnant lipase after the interaction with quercetin and EGCG. All the samples were analyzed in duplicate. The error bars represent the standard deviation. (C, D) UV fractograms for lipase quantification after the interaction with different concentrations of quercetin and EGCG.

The flavonoid concentrations chosen in this study are common in food products. For reference, the amount of quercetin and EGCG in green tea infusions can vary from 10 µM to 8000 µM, according to Phenol-Explorer database [32]. Considering a median of 4000 µM flavonoid concentration and following the dissolution process suggested by the INFOGEST protocol, the concentration of flavonoids in the small intestine would be 500 µM, suggesting that these flavonoids could indeed be present as aggregates in the small intestine.

## 4. Conclusions

The properties of quercetin and EGCG aggregates under simplified intestinal conditions were investigated. Both, quercetin and EGCG precipitate in aqueous solutions similar to those found in the small intestine. When interacting in a salt solution, they are affected in different

ways, quercetin aggregates are stabilized and EGCG aggregates are destabilized. Despite their different behavior in the salt solution, both of them interact with pancreatic lipase resulting in a sequestering of the enzyme. The results show that colloidal particles of aggregated flavonoids can affect lipase in solution. The results give an initial indication of the mechanism of inhibition between flavonoid aggregates and pancreatic lipase. As discussed in the previous section, it is likely that these aggregates can be present in the small intestine of humans. Therefore, comprehension of the interaction between EGCG or quercetin and lipase as well as the mechanism of inhibition is important to understand the effect of EGCG and quercetin in human diets.

## Supporting information

**S1 Fig. SDS-PAGE for porcine pancreatic lipase.**
(DOCX)

## Author Contributions

**Conceptualization:** Atma-Sol Bustos, Andreas Håkansson, Javier A. Linares-Pastén, Lars Nilsson.

**Formal analysis:** Andreas Håkansson.

**Funding acquisition:** J. Mauricio. Peñarrieta, Lars Nilsson.

**Investigation:** Atma-Sol Bustos.

**Methodology:** Atma-Sol Bustos.

**Project administration:** Lars Nilsson.

**Resources:** Andreas Håkansson, Javier A. Linares-Pastén, J. Mauricio. Peñarrieta, Lars Nilsson.

**Supervision:** J. Mauricio. Peñarrieta, Lars Nilsson.

**Validation:** Atma-Sol Bustos, Javier A. Linares-Pastén.

**Visualization:** Atma-Sol Bustos, Andreas Håkansson, Javier A. Linares-Pastén, J. Mauricio. Peñarrieta, Lars Nilsson.

**Writing – original draft:** Atma-Sol Bustos.

**Writing – review & editing:** Andreas Håkansson, Javier A. Linares-Pastén, J. Mauricio. Peñarrieta, Lars Nilsson.

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
