## [Decision Letter · Decision Letter 0]

13 Feb 2020

PONE-D-19-29298

INTERACTION OF QUERCETIN AND EPIGALLOCATECHIN GALLATE (EGCG) AGGREGATES WITH PANCREATIC LIPASE UNDER SIMPLIFIED INTESTINAL CONDITIONS

PLOS ONE

Dear Ms Bustos,

Thank you for submitting your manuscript to PLOS ONE. After careful consideration, we feel that it has merit but does not fully meet PLOS ONE’s publication criteria as it currently stands. Therefore, we invite you to submit a revised version of the manuscript that addresses the points raised during the review process.

We would appreciate receiving your revised manuscript by Mar 29 2020 11:59PM. To enhance the reproducibility of your results, we recommend that if applicable you deposit your laboratory protocols in protocols.io, where a protocol can be assigned its own identifier (DOI) such that it can be cited independently in the future. For instructions see: http://journals.plos.org/plosone/s/submission-guidelines#loc-laboratory-protocols

We look forward to receiving your revised manuscript.

Kind regards,

Oliver Chen

Academic Editor

PLOS ONE

Journal Requirements:

2. To comply with PLOS ONE submission guidelines, in your Methods section, please provide additional information regarding your statistical analyses to ensure the reproducibility of your analysis (e.g. include the name and version of any software used, which t-test was used and any procedures required to reproduce the analysis). For more information on PLOS ONE's expectations for statistical reporting, please see https://journals.plos.org/plosone/s/submission-guidelines.#loc-statistical-reporting.

Reviewers' comments:

Reviewer's Responses to Questions

**Comments to the Author**

1. Is the manuscript technically sound, and do the data support the conclusions?

Reviewer #1: Partly

2. Has the statistical analysis been performed appropriately and rigorously? 

Reviewer #1: Yes

3. Have the authors made all data underlying the findings in their manuscript fully available?

Reviewer #1: Yes

4. Is the manuscript presented in an intelligible fashion and written in standard English?

Reviewer #1: Yes

5. Review Comments to the Author

Reviewer #1: This manuscript describes a set of studies examining the interaction of quercetin and EGCG in a simplified intestinal model. Overall, the manuscript is written well and the literature cited is appropriate. There are a couple of general concerns with the manuscript and several specific points listed below. The first general concern is that the authors need to establish whether the concentrations of quercetin and EGCG are biologically relevant within their model. What is the expected concentration of these compounds in the intestine? The second concern is with the conclusion. How are these aggregates important, or not, within their model? How might this affect a person/animal taking these compounds at the concentrations given relevant?

1. Line 40, reference 10: This reference does not appear to be complete.

2. Lines 62-64: The part beginning with "Dimethyl..." is not a sentence.

3. Line 66 and 68: References 22 and 23 and the same in the reference section.

4. Line 73: Why are the solutions prepared in DMSO? The flavonoids are not taken dissolved in such a solvent.

5. Lines 134-136: The authors should consider multiply the EGCG response by a factor to make it easier to see on the plot.

6. Lines 161-164: Are the precipitates proteins or particle (or both)? The same terminology should be used throughout the manuscript.

6. PLOS authors have the option to publish the peer review history of their article (what does this mean?). If published, this will include your full peer review and any attached files.

Reviewer #1: No

---

## [Author Response · Author response to Decision Letter 0]

23 Mar 2020

Dear reviewers, 

We highly appreciate all the comments and we agree that we need to provide more information. Please see all the answers below. 

Reviewers' Comments to the Author:

Reviewer #1

This manuscript describes a set of studies examining the interaction of quercetin and EGCG in a simplified intestinal model. Overall, the manuscript is written well and the literature cited is appropriate. There are a couple of general concerns with the manuscript and several specific points listed below. The first general concern is that the authors need to establish whether the concentrations of quercetin and EGCG are biologically relevant within their model. What is the expected concentration of these compounds in the intestine? The second concern is with the conclusion. How are these aggregates important, or not, within their model? How might this affect a person/animal taking these compounds at the concentrations given relevant?

Answer for the general concerns: 

The flavonoid concentration in the intestine was now estimated using INFOGEST protocol and Phenol-Explorer database, see lines 238-240. Some comments in the conclusion section were also added, see lines 247-250.

1. Line 40, reference 10: This reference does not appear to be complete.

Answer: The reference has been corrected, see line 276.

2. Lines 62-64: The part beginning with "Dimethyl..." is not a sentence.

Answer: The correction has been made, see line 64.

3. Line 66 and 68: References 22 and 23 and the same in the reference section.

Answer: Reference 23 was deleted and the rest of the references were adjusted according to it. See line 69. 

4. Line 73: Why are the solutions prepared in DMSO? The flavonoids are not taken dissolved in such a solvent.

Answer: The solubility of the flavonoids presents a considerable experimental challenge. Thus, in order to be able to make quantitative studies it is vital to have control of a least the initial concentrations. This requires initial dissolution of the flavonoids and for this purpose DMSO is used. In the aqueous solution/dispersion during digestion, the concentration of flavonoids is in the range of what has been reported for instance for green tea infusions (see lines 237-239). Hence, the resulting concentrations are in a relevant range. The final solution concentration of DMSO is 2.5%. For solvent properties it is the volume fraction that will be the more relevant concentration and this will be slightly lower than 2.5% as DMSO has a higher density than water. Thus, the influence of DMSO on solvent properties can be considered negligible.

5. Lines 134-136: The authors should consider multiply the EGCG response by a factor to make it easier to see on the plot.

Answer: The values of the plot are now multiplied by 7 in order to improve the visualization of the plot. See the attached figure “Fig 1 corrected” and lines 136 – 138 for clarifications.

6. Lines 161-164: Are the precipitates proteins or particle (or both)? The same terminology should be used throughout the manuscript.

Answer: The paragraph has been modified for better understanding, see lines 165 – 168. The terminology was checked along the whole manuscript.

---

## [Editor Report · Decision Letter 1]

30 Mar 2020

INTERACTION OF QUERCETIN AND EPIGALLOCATECHIN GALLATE (EGCG) AGGREGATES WITH PANCREATIC LIPASE UNDER SIMPLIFIED INTESTINAL CONDITIONS

PONE-D-19-29298R1

Dear Dr. Bustos,

We are pleased to inform you that your manuscript has been judged scientifically suitable for publication and will be formally accepted for publication once it complies with all outstanding technical requirements.

With kind regards,

Oliver Chen

Academic Editor

PLOS ONE
---

## [Editor Report · Acceptance letter]

2 Apr 2020

PONE-D-19-29298R1 

INTERACTION OF QUERCETIN AND EPIGALLOCATECHIN GALLATE (EGCG) AGGREGATES WITH PANCREATIC LIPASE UNDER SIMPLIFIED INTESTINAL CONDITIONS 

Dear Dr. Bustos:

I am pleased to inform you that your manuscript has been deemed suitable for publication in PLOS ONE. Congratulations! Your manuscript is now with our production department. 

With kind regards,

on behalf of

Dr. Oliver Chen 

Academic Editor

PLOS ONE